# Relevant Indicators of Consciousness After Head-Only Electrical Stunning in Rabbits, Stunning Efficiency, and Risk Factors in Commercial Conditions

**DOI:** 10.3390/ani15040587

**Published:** 2025-02-18

**Authors:** Alexandra Contreras-Jodar, Virginie Michel, Leonardo James Vinco, Aranzazu Varvaró-Porter, Antonio Velarde

**Affiliations:** 1Institut de Recerca i Tecnologia Agroalimentàries (IRTA), Animal Welfare, Veïnat de Sies, Monells, 17121 Girona, Spain; alexandra.contreras@irta.cat (A.C.-J.); aranzazu.varvaro@irta.cat (A.V.-P.); 2Direction of Strategy and Programmes, French Agency for Food, Environmental and Occupational Health & Safety (ANSES), 94701 Maisons-Alfort, France; virginie.michel@anses.fr; 3Istituto Zooprofilattico Sperimentale Della Lombardia e Dell’Emilia Romagna “Bruno Ubertini” (IZSLER), 25124 Brescia, Italy; leonardojames.vinco@izsler.it

**Keywords:** animal-based indicators, inter-observer repeatability, state of consciousness, stunning efficiency, head-only electrical stunning, rabbits, welfare, slaughterhouse

## Abstract

Head-only electrical stunning (HOES) is the predominant method for stunning rabbits in commercial slaughterhouses. It carries the risk of ineffective stunning and, as a reversible stunning method, regaining consciousness before death through exsanguination. Operators must be trained to recognise signs of consciousness and reapply stunning if necessary to minimise pain and suffering. However, the lack of a standardised list of indicators for assessing consciousness results in variability in evaluations. While the European Food Safety Authority (EFSA) has evaluated the validity and feasibility of such indicators, their repeatability remains underexplored. This study aims to assess the repeatability of valid and feasible consciousness indicators to evaluate the efficiency of stunning, with the goal of proposing a refined list of indicators for use in commercial slaughterhouses. The findings offer insights for improving animal welfare standards and developing standardised monitoring protocols for consciousness assessment.

## 1. Introduction

Currently, approximately 62 million rabbits are slaughtered annually in approved slaughterhouses across the European Union (EU) [1]. The EU regulation on animal welfare at slaughter requires that animals must be rendered unconscious through stunning prior to slaughter, ensuring they are insensible to pain until death occurs [2].

Regular inspections are required to assess the effectiveness of stunning, and if an animal is found to have failed to be rendered unconscious or to have regained consciousness, immediate corrective action must be taken to prevent unnecessary suffering [2]. Five methods of rabbit stunning are permitted: penetrative and non-penetrative captive bolt, percussive blow to the head, head-only electrical stunning (HOES), and head-to-body electrical stunning. A 2022 survey revealed that 93% of EU slaughterhouses use HOES, with 99.99% of rabbits stunned using this method, while 7% use captive bolt [2].

The HOES devices can be either fixed or mobile. Fixed devices involve guiding the rabbit’s head into V-shaped stunning tongs, while mobile devices require operators to position the tongs on the rabbit’s head. A variant of the fixed device includes a channel-shaped support to aid head positioning [3]. In any case, HOES involves applying a sufficient electrical current to induce a generalised epileptic seizure, rendering rabbits temporarily unconscious [4,5].

Although the minimum electrical parameters for stunning other species are legally defined, no specific requirements exist for rabbits in EU regulations [2]. The European Commission compiled the thresholds found in different national guidelines aimed at providing recommendations [6]. The minimum current is provided in a range between 140 and 400 mA per rabbit, with higher currents increasing the likelihood of effective stunning. While no maximum frequency is specified, EFSA recommends 50 Hz as the most common frequency in commercial settings, noting that lower frequencies are more effective. A shorter stun-to-stick interval is also preferred to minimise the risk of regaining consciousness [3].

Monitoring the state of consciousness after stunning is mandatory [2], and backup stunning methods must be applied if ABIs indicate consciousness.

ABIs must be valid, feasible, and repeatable to be relevant (and effectively assess the state of consciousness). Validity reflects how accurately an indicator provides meaningful information about the presence of conscious birds. Feasibility refers to its adaptability across different slaughterhouse (SH) configurations, and processing speeds. Repeatability measures the consistency of results, either when the same observer conducts repeated assessments (intra-observer repeatability) or when multiple trained observers achieve similar outcomes (inter-observer repeatability—IOR). If the repeatability is low, the indicator is likely unsuitable for reliable welfare assessments.

In 2020, EFSA compiled a list of indicators of the state of consciousness in head-only electrically stunned rabbits reported in the scientific literature and ranked the indicators according to the validity (i.e., sensitivity) and feasibility [3]. The indicators with the highest validity and feasibility were designated as recommended ABIs, while the others were categorised as optional. However, information on the IOR and which ABIs are the most observed in rabbits remained a knowledge gap.

A survey conducted across EU rabbit SHs revealed considerable variability in the electrical key parameters applied, as well as a lack of uniformity in the indicators used to monitor the state of consciousness following HOES [1].

This study aims to provide a refined list of ABIs to be used by official inspectors and business operators for assessing the state of consciousness in rabbits after HOES in commercial SHs. For this, the IOR of the recommended ABIs identified by EFSA [3] was assessed across different commercial SHs and batches of rabbits. Additionally, the prevalence of the outcomes of consciousness for each ABI was calculated to identify those most likely to occur in instances of ineffective stunning, and correlations between the ABIs were explored. Based on these findings, a streamlined list of relevant ABIs is proposed for assessing the state of consciousness in rabbits after HOES in commercial SHs. Moreover, it will assess the efficiency of stunning using different combinations of key parameters and identify factors that contribute to effective stunning. Findings are intended to directly aid in the implementation of standardised monitoring protocols and improve welfare standards.

## 2. Materials and Methods

### 2.1. Selection of Slaughterhouses and Animals

Sixteen commercial rabbit SHs equipped with head-only electrical stunners were selected in France (*n* = 5), Spain (*n* = 6), and Italy (*n* = 5), the three main countries producing rabbits in the EU [1]. A pre-selection of the SHs was carried out together with the official veterinary services of the corresponding Member States to reflect a certain diversity in terms of size of the SH, electrical key parameters, rabbit genotype, and line speed. The 16 SHs were chosen based on availability, as we could only include those that permitted our access to their facilities. No prior information regarding the stunning efficiency was provided. To maintain the anonymity of the SHs, each was assigned a unique identification number (1 to 16) in this study.

### 2.2. Description of the Slaughterhouses and Management

In all SHs, rabbits were head-only electrically stunned and afterwards shackled by their rear legs on the moving slaughter line. Substantial variations in the age, design, and layout of the SHs were observed, and the main management characteristics are described in Table 1. In 4 out of the 16 SHs, the rabbits were wet prior to stunning with the purpose of reducing electrical resistance caused by the fur and improving the stunning efficiency. Most SHs were equipped with more than one stunner used at the same time, and, in some cases, more than one operator oversaw bleeding. In these cases, the stun-to-stick interval differed depending on how far away each stunner was from the bleeder. SH-10 used a device that stunned and, immediately after, automatically cut the rabbit’s neck when deemed appropriate by the operator (i.e., after effective stunning assessed by tonic seizure of the animal). The bleeding procedure differed among SHs; seven SHs performed manual bleeding through ventral neck cut, seven of them carried out manual bleeding through lateral neck cut, and one of them (SH-10) performed automatic ventral neck cut with one operator checking that the neck cut was appropriately performed on all rabbits.

All stunners had a digital control panel that also automatically recorded the main electrical parameters applied (i.e., the actual total current passing through the rabbit’s head, voltage, and frequency) and the duration of exposure to the electrical tongs. These parameters were obtained from the official veterinary service and food business operators. A summary of the electrical parameters applied to each batch and SH, as well as the characteristics of the animals in each batch and the number of rabbits assessed, is provided in Table 2. The slaughter line speed ranged from 600 to 3600 rabbits/h.

### 2.3. Assessment of the State of Consciousness

#### 2.3.1. Observers

Four trained observers (A–D) assessed stunning effectiveness. Rabbits were randomly selected and highlighted with a laser pointer to ensure all observers evaluated the same animals. Stunning effectiveness was assessed on a representative sample of rabbits in each batch at two different stages along the slaughter line: (stage 1) immediately after stunning but before neck cutting, and (stage 2) during bleeding (see Figure 1).

The observers individually scored the ABIs within specific time frames that varied due to the SH design and visibility of the rabbits. In stage 1, the assessment was performed 2 to 10 s post-stunning (see Figure 2). In stage 2, the observers were placed at a distance from the bleeder where they detected rabbits that began to show outcomes of consciousness and then, the rabbits were assessed for 6 to 15 s (Figure 3). Observers assessed the ABIs independently without discussing or sharing their assessments during the evaluation process.

In SH-3 and SH-10, it was not possible to assess the state of consciousness immediately after stunning (Figure 2). In SH-3, this was due to space constraints and in SH-10, it was because the stunner was equipped with an automatic neck-cutting device that operated immediately after stunning.

#### 2.3.2. Sample Assessment

All rabbit batches slaughtered during the observers’ presence in the SH were evaluated. For each batch, samples of 50 to 200 rabbits were assessed both before and during bleeding. This cycle was repeated until the entire batch was slaughtered to maximise the sample size.

Observers noted instances where distractions (e.g., business operators obstructing their view) prevented them from assessing a laser-indicated rabbit. In such cases, the observations of the other three observers were filtered out for repeatability analysis. A summary of the characteristics of the rabbits in the batch and the number of assessed rabbits is shown in Table 2. Although it is known that genetic factors influence stunning efficiency (differences in the amount of fur between genetics varies the resistance to electricity and therefore to the applied current), the impact of genetics could not be isolated due to confounding factors such as electrical parameters and batch weight.

#### 2.3.3. Indicators for the Assessment

The ABIs for the assessment of the state of consciousness immediately after stunning and during bleeding were selected based on those identified by EFSA as the most valid and feasible [5].

The selected ABIs immediately after stunning were tonic-clonic seizure, breathing, spontaneous blinking, and vocalisations, while those selected during bleeding were the same with the addition of righting reflex. Although EFSA proposed the use of corneal and palpebral reflexes at both stages, these were excluded from consideration due to feasibility concerns and impaired visibility, as these indicators require direct contact with the rabbit’s cornea or the inner/outer eye canthus or eyelashes. The description and outcomes associated with the consciousness and unconsciousness of these ABIs are outlined in Table 3. The four trained observers reached a consensus prior to the assessment on the indicators and the outcomes of consciousness and unconsciousness, the methodology of assessment, and the scoring system to standardise the protocol when assessing the rabbits.

**Figure 2 animals-15-00587-f002:**
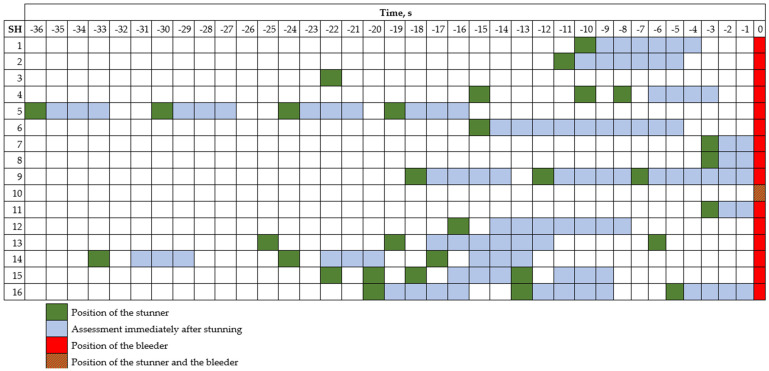
Position of the stunner(s) relative to the bleeder(s), expressed as the time interval between stunning and neck cutting, and the time frame during which the state of consciousness was assessed immediately after stunning and before bleeding, according to the slaughterhouse (SH) assessed.

**Figure 3 animals-15-00587-f003:**
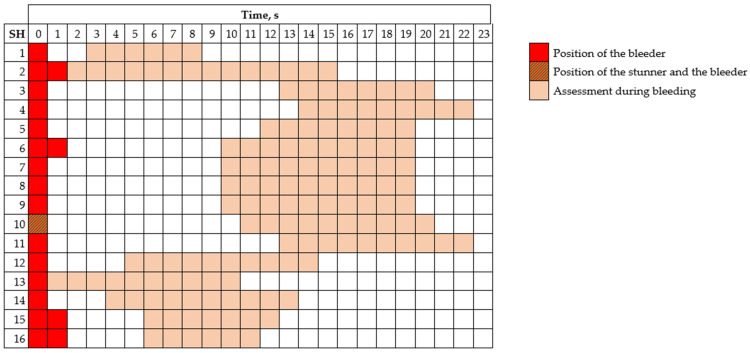
Position of the bleeder(s) and the time frame during which the state of consciousness was assessed during bleeding, according to the slaughterhouse (SH) assessed.

**Table 3 animals-15-00587-t003:** Description of the animal-based indicators (ABIs) and outcomes of unconsciousness and consciousness scores in head-only electrically stunned rabbits assessed in two different stages: immediately after stunning and during bleeding.

Stage	ABIs	Sign of Unconsciousness	Sign of Consciousness
Immediately after stunning and during bleeding	Tonic–clonic seizure	Rabbit shows arched and stiff neck (i.e., necks appear parallel to the ground), and paws and ears held tightly close to the body. Then, followed (or not) by kicking action and/or leg paddling that can be either rhythmic or erratic	General loss of muscle tone and a completely relaxed and flaccid body, with no neck tension.
	Breathing	Absence of opening of the mouth and thoracic or abdominal movements associated with cessation of inhalation and expiration. Presence of one movement is not considered as breathing.	Presence of rhythmic breathing considered as a minimum of two openings of the mouth and thoracic or abdominal muscles associated with inhalation and expiration with similar cadence.
	Spontaneous blinking	Rabbit does not open/close eyelid on its own (fast or slow) without stimulation.	Rabbit opens/closes eyelid on its own (fast or slow) without stimulation.
	Vocalisations	Absence of single or repeated short and loud shrieking (screaming).	Single or repeated shrieking (screaming).
During bleeding	Righting reflex	Absence of attempt to regain posture and/or raise the head.	Attempt to regain posture and/or raise the head.

The observers positioned themselves to obtain the best possible view of the shackled rabbits, typically from a ventrolateral angle. However, in some cases, due to variation in the design and construction of the SHs, the rabbits had to be assessed from a dorsolateral position instead of ventrolateral, both immediately after stunning and during bleeding (SH-2, SH-9, and SH-13), or immediately after stunning but not during bleeding (SH-15), or only during bleeding (SH-1, SH-4, SH-5, and SH-16). This impaired the assessment of breathing since the mouths were not clearly visible. Data were recorded in a binary format: 0 indicated an outcome of unconsciousness, and 1 indicated a sign of consciousness. The presence of at least one indicator with a sign of consciousness may indicate that the rabbit is conscious or gradually regaining consciousness after stunning, suggesting ineffective stunning or the recovery of consciousness.

### 2.4. Statistical Analysis

Data pre-processing, statistical analyses, and plotting were conducted using R software v.4.1.0. [7]. Initially, rabbits that were not assessed by all four observers were excluded to ensure the comparability of the observations. Statistical significance was set at *p* < 0.05 for all analyses.

#### 2.4.1. Inter-Observer Repeatability of Animal-Based Indicators

The overall level of agreement between observers for each ABI was determined and expressed by the crude proportion of agreement (PoA) and the Fleiss’ kappa (κ) using the “irr” package of R software [8]. The PoA can be misleading as it does not account for the scores that may result from random chance. κ overcomes this issue by providing a measure of inter-observer agreement when the assessed variable is binomial or categorical. It reflects the extent to which the observed PoA among observers exceeds what would be expected if ratings were made completely at random. κ ranges from −1 to +1, where 0 indicates the agreement expected from random chance, and 1 indicates perfect agreement between observers [9]. As a standardised value, κ is interpreted consistently across studies. According to Fleiss et al. [10], κ values greater than 0.75 indicate “excellent” agreement beyond chance, values between 0.40 and 0.75 indicate “fair to good” agreement, and values below 0.40 suggest “poor” agreement. However, when there is limited scoring variation in the evaluated indicator (i.e., low prevalence of indicators of the state of consciousness), κ may approach 0 despite high inter-observer agreement.

#### 2.4.2. Prevalence and Relationship Among Animal-Based Indicators

The chi-squared % defective test was used to assess whether there were statistically significant differences (divergence) among observers between the expected and observed frequencies for each sign of consciousness of the evaluated indicators. If one observer showed a statistically significant difference in their evaluation of ABIs compared to the others, the mean proportion of the closest evaluations or the intermediate value (when scoring was not consistent among observers) was recorded. Proportions among combinations of ABIs were visualised using Venn diagram, considering all rabbits assessed in the study using the “eulerr” package [11].

#### 2.4.3. Relationship Between Electrical Parameters and Stunning Efficiency

The stunning inefficiency of each batch and SH was evaluated by calculating the percentage of rabbits displaying at least one sign of consciousness at any stage of the assessment: immediately after stunning and during bleeding. Chi-squared % defective test was used to identify any statistical differences among observers between the expected and the observed frequencies for each outcome of the evaluated indicators. If an observer’s evaluation differed significantly from the others, the mean proportion of the two closest evaluations or the intermediate value (when scoring was inconsistent among observers) was reported.

The prevalence of the sign of consciousness for each indicator within each batch was calculated and, from this, the interval of confidence for the population (i.e., each batch) was calculated. A 95% confidence interval for rabbits showing outcomes of consciousness was computed using the Wilson’s formula, available in the “epitools” package of R software [12].

#### 2.4.4. Factors That Influence the Stunning Efficiency

Different variables were explored to check if they could be risk factors associated with inefficient stunning. The following variables, wetting the rabbit’s head, the electrical stunning parameters, and the stun-to-stick interval (s), were evaluated using binomial logistic regression with the “MASS” package [13].

For this, data were converted to a binary format. For every rabbit assessed, a 0 was recorded when no indicators of consciousness were detected, and a 1 was recorded if at least one indicator of consciousness was observed. Then, a rabbit was classified as inefficiently stunned (annotated as 1) if at least two observers recorded it as having at least one sign of consciousness; otherwise, it was classified as efficiently stunned (annotated as 0).

The variables were transformed into binary or multinomial format: rabbits slaughtered in SHs that wet their heads before stunning were coded as 1; those that did not were coded as 0. For the electrical parameters, four groups were initially defined based on the combination of current and frequency. Various current thresholds were tested (140, 200, and 300 mA). Rabbits stunned with currents below these thresholds and a frequency of 50 Hz were categorised as “low current-low frequency”. Conversely, those stunned with currents above the thresholds at 50 Hz were classified as “high current-low frequency.” Rabbits exposed to both higher currents and frequencies above 50 Hz were placed in the “high current-high frequency” group. Those stunned with lower currents, but higher frequencies were intended to be labelled as “low current-high frequency.” However, no rabbits fell into this last category, so the model ultimately included only three electrical parameter groups. Stun-to-stick intervals of up to 5 s were coded as 1, and those over 5 s were coded as 0.

Model selection was based on the Akaike Information Criterion [14]. Then, the variables included in the model were checked for multicollinearity using the “car” package [15]. Multicollinearity is considered negligible with variance inflation factors (VIFs) below 5 [16]. Checking for multicollinearity ensures stable, interpretable, and reliable coefficients. Odds ratios (OR) with 95% confidence intervals (95% CI) were used to express coefficients. An OR greater than 1 with a 95% CI excluding 1, indicates that the factor has an “adverse” effect, decreasing the odds of efficient stunning. Conversely, an OR less than 1 with a 95% CI excluding 1 indicated that the factor has a “protective” effect, increasing the odds of efficient stunning.

## 3. Results

The ABIs were assessed on a total of 4112 rabbits immediately after stunning and 7428 during bleeding from 16 different SHs across France, Spain, and Italy by four observers (A, B, C, and D).

### 3.1. Inter-Observer Repeatability of the Animal-Based Indicators

#### 3.1.1. Immediately After Stunning

Immediately after stunning, in all SHs but SH-3 and SH-10, four ABIs of the state consciousness were assessed: tonic–clonic seizure, breathing, spontaneous blinking, and vocalisation. The average prevalence of rabbits between batches showing at least one indicator, by observer and SH, is shown in Table 4. On the other hand, the overall level of agreement between the four observers for these ABIs according to the SH is shown in Table 5.

##### Tonic–Clonic Seizure

Some rabbits showed an absence of tonic–clonic seizure after stunning in six out of 16 SHs. The highest prevalence in a sample was found in SH-5 (27/183; 14.8%) followed by SH-4 (9/199; 4.5%), SH-9 (2/154; 1.3%), SH-16 (1/200; 0.3%), SH-14 (1/400; 0.3%), and SH1 (1/491; 0.002%) as shown in Table 4. Observers did not differ significantly in the detecting a lack of tonic–clonic seizure in any of the SHs evaluated (*p* > 0.05). The PoA was above 91.3% in all the SHs and the κ and its interpretation strongly differed between SHs ranging from “poor” to “excellent” agreement (Table 5). The Fleiss’ kappa coefficient could not be computed neither in SH-6 and SH-11 nor in SH-13 due to the absence of scoring variation as all rabbits assessed showed the presence of tonic–clonic seizure (Table 5).

Considering the data from the total of rabbits assessed in the present study at this stage, a mean of 1.1% (n = 4112) of the rabbits showed the absence of tonic–clonic seizure with a similar prevalence between observers (*p* > 0.05). The PoA among observers was 98.5% and the κ was statistically significant and interpreted as “fair to good” (*p* < 0.001; κ = 0.67; Table 5).

**Table 4 animals-15-00587-t004:** Number of rabbits with outcomes of consciousness immediately after head-only stunning according to the observer (A to D) and the slaughterhouse (SH) evaluated.

		Absence of TC, %						Presence of BR, %						Presence of SB, %						Presence of VC, %					
SH	n	A	B	C	D	Mean	*p*-Value	A	B	C	D	Mean	*p*-Value	A	B	C	D	Mean	*p*-Value	A	B	C	D	Mean	*p*-Value
1	491	4	3	4	0	3	0.265	25	20	27	25	24	0.634	26	21	29	26	26	0.714	0	0	0	0	0	-
2	333	0	0	0	1	0	0.388	0	0	0	0	0	-	0	0	0	0	0	-	0	0	0	0	0	-
3	0	-	-	-	-	-	-	-	-	-	-	-	-	-	-	-	-	-	-	-	-	-	-	-	-
4	199	10	10	8	8	9	0.924	0	0	0	0	0	-	5	5	6	4	5	0.938	0	0	0	0	0	-
5	183	24	29	27	26	27	0.902	0	0	0	0	0	-	0	0	0	0	0	-	0	0	0	0	0	-
6	276	0	0	0	0	0	-	0	0	0	0	0	-	0	0	0	0	0	-	0	0	0	0	0	-
7	381	0	0	1	0	0	0.391	0	0	0	0	0	-	0	0	0	0	0	-	0	0	0	0	0	-
8	200	0	0	1	0	0	0.391	0	1	0	0	0	0.391	0	0	0	0	0	-	0	0	0	0	0	-
9	154	2	3	2	2	2	0.953	0	0	0	0	0	-	0	1	0	0	0	0.391	0	0	0	0	0	-
10	0	-	-	-	-	-	-	-	-	-	-	-	-	-	-	-	-	-	-	-	-	-	-	-	-
11	400	1	0	0	0	0	0.391	0	0	0	0	0	-	0	0	0	0	0	-	0	0	0	0	0	-
12	197	0	0	0	0	0	-	1	0	2	0	1	0.298	1	0	2	0	1	0.298	0	0	0	0	0	-
13	398	0	0	0	0	0	-	0	0	0	0	0	-	0	0	0	0	0	-	0	0	0	0	0	-
14	400	1	1	1	2	1	0.896	0	0	0	0	0	-	0	0	0	0	0	-	0	0	0	0	0	-
15	200	0	0	1	0	0	0.391	0	0	0	0	0	-	0	0	0	0	0	-	0	0	0	0	0	-
16	300	1	1	1	1	1	1.000	0	0	0	0	0	-	0	0	0	0	0	-	0	0	0	0	0	-
All	4112	43	47	46	40	44	0.296	26	21	29	25	25	0.728	27	22	31	26	27	0.503	0	0	0	0	0	-

TC: tonic–clonic seizure; BR: breathing; SB: spontaneous blinking; VC: vocalisation; n: number of total rabbits assessed.

**Table 5 animals-15-00587-t005:** Inter-observer proportion of agreement (PoA), Fleiss’ kappa coefficient (κ), and interpretation of the animal-based indicators for the state of consciousness of head-only electrically stunned rabbits assessed immediately after stunning according to the slaughterhouse assessed.

		Tonic-Clonic Seizure			Breathing			Spontaneous Blinking			Vocalisation		
SH	n	PoA, %	κ	*p*-Value	PoA, %	κ	*p*-Value	PoA, %	κ	*p*-Value	PoA, %	κ	*p*-Value
1	491	94.7	0.127 (P)	<0.001	93.7	0.635 (FG)	<0.001	98.4	0.273 (P)	<0.001	100	*	*
2	333	99.7	0.000 (P)	0.973	100	*	*	98.8	0.130 (P)	<0.001	100	*	*
3	0	-	-	-	-	-	-	-	-	-	-	-	-
4	199	96.5	0.767 (E)	<0.001	100	*	*	96.5	0.590 (FG)	<0.001	100	*	*
5	183	91.3	0.816 (E)	<0.001	100	*	*	100	*	*	100	*	*
6	276	100	*	*	100	*	*	100	*	*	100	*	*
7	381	99.7	0.000 (P)	0.975	100	*	*	100	*	*	100	*	*
8	200	99.5	−0.001 (P)	0.965	100	*	*	100	*	*	100	*	*
9	154	99.4	0.887 (E)	<0.001	99.4	−0.002 (P)	0.961	100	*	*	100	*	*
10	0	-	-	-	-	-	-	-	-	-	-	-	-
11	200	100	*	*	100	*	*	100	*	*	100	*	*
12	397	99.7	−0.001 (P)		99.5	0.221 (P)		99.2	0.165 (P)	<0.001	100	*	*
13	398	100	*	*	100	*	*	100	*	*	100	*	*
14	400	99.8	0.799 (E)	<0.001	100	*	*	100	*	*	100	*	*
15	200	99.5	−0.001 (P)	0.965	100	*	*	100	*	*	100	*	*
16	300	97.7	0.358 (P)	<0.001	100	*	*	100	*	*	100	*	*
All	4112	98.5	0.669 (FG)	<0.001	99.2	0.631 (FG)	<0.001	99.5	0.405 (FG)	<0.001	100	*	*

* Insufficient scoring variation to calculate kappa coefficients (all indicator scores were 0). Kappa interpretation: ≥0.75 ‘excellent’ (E), 0.40–0.74 ‘fair to good’ (FG), and <0.40 ‘poor’ agreement (P) [10].

##### Breathing

Rabbits showing signs of breathing were observed in 2 out of 16 SHs. The highest prevalence of breathing in a sample was found in SH-1 (24/491; 4.9%) followed by SH-12 (1/197; 0.5%) as shown in Table 4. Similarly, as in the absence of tonic–clonic seizure, the observers did not differ significantly in the detected prevalence of breathing in any of the SHs evaluated (*p* > 0.05). The PoA between observers was above 93.7% in all SHs and there was divergence in κ linked to different degrees in signs of breathing among SHs ranging from “poor” to “fair to good” agreement (Table 5).

Taking into consideration all rabbits from the SHs assessed, a mean of 0.6% (n = 4112) of the animals showed signs of breathing before neck cutting with a similar prevalence between observers (*p* > 0.05) as shown in Table 4. The PoA among observers was 99.2% and the κ was statistically significant and interpreted as “fair to good” (*p* < 0.001; κ = 0.63; Table 5).

##### Spontaneous Blinking

Rabbits with spontaneous blinking were observed in 3 out of 16 SHs. The highest prevalence was found in SH-1 (26/491; 5.3%) followed by SH-4 (5/199; 2.5%) and SH-12 (1/197; 0.5%) as reported in Table 4.

Again, the observers did not differ significantly in the detected prevalence of presence of spontaneous blinking in any of the SHs evaluated (*p* > 0.05). The PoA was above 96.5% in all SHs and there was divergence in κ among SHs ranging from “poor” to “fair to good” agreement (Table 5).

Taking into consideration all rabbits from the SHs assessed, a prevalence of 0.7% of the rabbits showed spontaneous blinking with a similar prevalence noted between observers (*p* > 0.05) as shown in Table 4. The PoA among observers was 99.5% and the κ was statistically significant and interpreted as “fair to good” (*p* < 0.001; κ = 0.41; Table 5).

##### Vocalisation

No vocalisation was heard in any of the SHs. Thus, the PoA was 100% in all SHs and the κ could not be computed.

#### 3.1.2. During Bleeding

Five ABIs were evaluated during bleeding: tonic–clonic seizure, breathing, spontaneous blinking, vocalisation, and righting reflex. The prevalence of rabbits showing outcomes of consciousness by observer and SH is shown in Table 6. The overall level of agreement between the four observers according to the SH is reported in Table 7.

##### Tonic–Clonic Seizure

Rabbits with absence of tonic–clonic seizure during bleeding were observed in all SHs and the highest prevalence was found in SH-8 (89.9%), while the lowest in SH-2 (5.7%) as shown in Table 6. There was uniformity in rating among the observers in only 8 out of the 16 SHs (*p* > 0.05; Table 7). The PoA ranged from 54.8 to 90.1% and the κ ranged from 0.44 to 0.72 being interpreted as “fair to good” in all SHs (Table 7).

Taking into consideration all rabbits from the SHs assessed, the prevalence of the absence of tonic–clonic seizure was 56.3% and the prevalence according to the observers differed statistically (*p* < 0.001; Table 6) since Obs-A noticed around 6% less rabbits than Obs-B and Obs-C. Furthermore, the PoA among observers was 72.9% and the κ was statistically significant and interpreted as “fair to good” agreement among observers (*p* < 0.001; κ = 0.70; Table 7).

**Table 6 animals-15-00587-t006:** Number of rabbits with outcomes of consciousness after bleeding in head-only stunned according to the observer (A to D) and the slaughterhouse (SH) evaluated.

		Absence of TC, %								Presence of BR, %									Presence of SB, %							
SH	n	A	B		C	D	Mean		*p*-Value	A	B		C	D	Mean		*p*-Value		A	B		C	D	Mean		*p*-Value
1	166	23	29		28	26	27		0.815	57 ^b^	68 ^b^		58 ^b^	24 ^a^	61		<0.001		55 ^b^	67 ^b^		58 ^b^	23 ^a^	60		<0.001
2	436	20	21		33	25	25		0.213	92	96		100	75	91		0.168		85 ^ab^	96 ^b^		95 ^ab^	66 ^a^	92		0.038
3	398	342	320		337	339	335		0.139	293 ^b^	276 ^ab^		288 ^b^	249 ^a^	286		0.003		296 ^b^	281 ^ab^		291 ^ab^	257 ^a^	289		0.012
4	633	491 ^ab^	510 ^b^		534 ^c^	451 ^a^	501		<0.001	364 ^b^	341 ^b^		379 ^b^	268 ^a^	361		<0.001		360 ^b^	341 ^b^		379 ^b^	262 ^a^	360		<0.001
5	380	225 ^ab^	260 ^c^		238 ^b^	200 ^a^	232		<0.001	211 ^b^	194 ^ab^		200 ^ab^	172 ^a^	202		0.036		214 ^b^	205 ^ab^		204 ^ab^	171 ^a^	208		0.010
6	294	100	83		85	90	90		0.427	55 ^b^	61 ^b^		57 ^b^	29 ^a^	58		0.002		84 ^b^	86 ^b^		80 ^b^	49 ^a^	83		<0.001
7	574	509	509		514	513	511		0.946	38	47		39	31	39		0.313		37	44		37	30	37		0.418
8	375	336	335		339	338	337		0.961	62	53		51	46	53		0.400		25	25		21	19	23		0.735
9	395	312 ^b^	303 ^ab^		321 ^b^	276 ^a^	312		<0.001	136 ^b^	130 ^ab^		131 ^b^	97 ^a^	132		0.010		173 ^b^	161 ^b^		163 ^b^	124 ^a^	166		0.002
10	588	366 a	448 ^b^		349 ^a^	386 ^a^	367		<0.001	19 ^ab^	34 ^b^		28 ^ab^	17 ^a^	27		0.045		11	19		16	8	14		0.137
11	548	470	467		485	475	474		0.403	72	64		71	59	67		0.586		20	21		29	25	24		0.525
12	461	136 ^a^	187 ^b^		178 ^b^	211 ^b^	192		<0.001	191 ^b^	181 ^b^		190 ^b^	142 ^a^	187		0.002		180 ^b^	156 ^ab^		200 ^c^	134 ^a^	190		<0.001
13	555	98 ^a^	136 ^b^		147 ^b^	140 ^b^	141		0.002	228 ^ab^	256 ^b^		220 ^ab^	211 ^a^	235		0.038		219	231		206	194	213		0.118
14	575	256 ^a^	321 ^b^		334 ^b^	299 ^ab^	318		<0.001	190 ^ab^	264 ^b^		229 ^b^	178 ^a^	228		<0.001		252 ^ab^	344 ^c^		282 ^b^	234 ^a^	267		<0.001
15	400	94 ^a^	103 ^a^		139 ^b^	127 ^ab^	108		<0.001	32 ^b^	28 ^ab^		14 ^a^	15 ^ab^	25		0.008		39	38		23	25	31		0.061
16	650	203	184		186	192	191		0.655	46 ^ab^	71 ^b^		50 ^ab^	39 ^a^	56		0.007		40 ^ab^	58 ^b^		35 ^ab^	30 ^a^	44		0.008
All	7428	3981 ^a^	4216 ^b^		4247 ^b^	4088 ^ab^	4184		<0.001	2086 ^b^	2164 ^b^		2105 ^b^	1652 ^a^	2108		<0.001		2090 ^b^	2164 ^b^		2119 ^b^	1651 ^a^	2116		<0.001
		**Presence of VC, %**												**Presence of RR, %**												
**SH**	**n**	**A**		**B**		**C**		**D**		**Mean**		** *p* ** **-Value**		**A**		**B**		**C**			**D**		**Mean**		** *p* ** **-Value**	
1	166	0		0		0		0		0		-		7		5		4			7		6		0.749	
2	436	0		0		0		0		0		-		3		5		2			6		4		0.471	
3	398	1		0		1		0		1		0.572		43		33		29			35		35		0.354	
4	633	0		0		0		0		0		-		37		37		47			28		37		0.161	
5	380	0		0		0		0		0		-		50 ^b^		37 ^b^		52 ^b^			18 ^a^		46		<0.001	
6	294	4		2		1		1		2		0.388		5 ^c^		1 ^b^		0 ^a^			0 ^a^		0		0.010	
7	574	0		0		0		0		0		-		0		0		0			0		0		-	
8	375	0		0		0		0		0		-		4		2		6			3		4		0.502	
9	395	0		0		2		0		0		0.111		13 ^ab^		15 ^ab^		28 ^b^			8 ^a^		12		0.003	
10	588	0		0		0		0		0		-		1		0		0			0		0		0.391	
11	548	1		0		1		1		1		0.801		4		3		7			3		4		0.466	
12	461	2		2		0		3		2		0.436		3		7		4			9		6		0.261	
13	555	0		0		0		0		0		-		0		2		4			1		2		0.171	
14	575	3		0		1		3		2		0.276		8 ^a^		37 ^b^		42 ^b^			24 ^b^		34		<0.001	
15	400	0 ^a^		0 ^a^		3 ^b^		0 ^a^		0		0.029		11		11		13			16		13		0.716	
16	650	1		0		5		2		2		0.071		47 ^b^		28 ^ab^		29 ^ab^			18 ^a^		25		0.002	
All	7428	12		4		14		10		10		0.132		236 ^b^		223 ^ab^		267 ^b^			176 ^a^		242		<0.001	

TC: tonic–clonic seizure; BR: breathing; SB: spontaneous blinking; n: number of total rabbits assessed; VC: vocalisation; RR: righting reflex; a–c  = values with different superscripts within the same raw differ among observers by chance (*p*  <  0.05).

**Table 7 animals-15-00587-t007:** Inter-observer proportion of agreement (PoA), Fleiss’ kappa coefficient (κ), and interpretation of the animal-based indicators for the state of consciousness of head-only electrically stunned rabbits assessed after bleeding according to the slaughterhouse assessed.

		Tonic-Clonic Seizure			Breathing			Spontaneous Blinking			Vocalisations			Righting Reflex		
SH	n	PoA, %	κ	*p*-Value	PoA, %	κ	*p*-Value	PoA, %	κ	*p*-Value	PoA, %	κ	*p*-Value	PoA, %	κ	*p*-Value
1	166	74.7	0.491 (FG)	<0.001	56.6	0.432 (FG)	<0.001	86.7	0.522 (FG)	<0.001	100	*	*	94.0	0.531 (FG)	<0.001
2	436	87.8	0.436 (FG)	<0.001	79.1	0.628 (FG)	<0.001	94.3	0.569 (FG)	<0.001	100	*	*	98.4	0.484 (FG)	<0.001
3	398	81.4	0.628 (FG)	<0.001	75.1	0.670 (FG)	<0.001	97.7	0.279 (P)	<0.001	99.7	0.332 (P)	<0.001	86.7	0.543 (FG)	<0.001
4	633	65.9	0.454 (FG)	<0.001	68.6	0.658 (FG)	<0.001	95.3	0.476 (FG)	<0.001	100	*	*	91.0	0.544 (FG)	<0.001
5	387	54.8	0.485 (FG)	<0.001	68.2	0.651 (FG)	<0.001	99.2	0.581 (FG)	<0.001	100	*	*	79.1	0.355 (FG)	<0.001
6	294	64.3	0.556 (FG)	<0.001	61.2	0.434 (FG)	<0.001	94.9	0.305 (P)	<0.001	98.0	0.090 (P)	<0.001	98.0	−0.005 (P)	0.829
7	574	90.1	0.718 (FG)	<0.001	96.2	0.827 (E)	<0.001	99.7	0.666 (FG)	<0.001	100	*	*	100	*	*
8	375	89.1	0.658 (FG)	<0.001	96.0	0.811 (E)	<0.001	97.9	0.375 (P)	<0.001	100	*	*	98.7	0.663 (FG)	<0.001
9	395	62.8	0.426 (FG)	<0.001	69.1	0.644 (FG)	<0.001	91.9	0.277 (P)	<0.001	99.5	−0.001 (P)	0.951	90.1	0.316 (P)	<0.001
10	588	61.7	0.500 (FG)	<0.001	97.8	0.728 (FG)	<0.001	99.3	0.531 (FG)	<0.001	100	*	*	99.8	0.000 (P)	0.980
11	548	79.0	0.524 (FG)	<0.001	92.5	0.807 (E)	<0.001	87.2	0.623 (FG)	<0.001	99.6	0.666 (FG)	<0.001	96.7	0.392 (P)	<0.001
12	461	60.3	0.533 (FG)	<0.001	60.1	0.519 (FG)	<0.001	86.1	0.488 (FG)	<0.001	99.1	0.379 (P)	<0.001	95.9	0.111 (P)	<0.001
13	555	74.1	0.606 (FG)	<0.001	68.8	0.629 (FG)	<0.001	95.3	0.418 (FG)	<0.001	100	*	*	98.7	−0.003 (P)	0.855
14	575	67.1	0.642 (FG)	<0.001	52.3	0.479 (FG)	<0.001	76.2	0.461 (FG)	<0.001	99.5	0.475 (FG)	<0.001	90.6	0.428 (FG)	<0.001
15	400	67.5	0.577 (FG)	<0.001	86.2	0.495 (FG)	<0.001	75.8	0.516 (FG)	<0.001	99.2	−0.002 (P)	0.927	92.5	0.359 (P)	<0.001
16	650	78.6	0.724 (FG)	<0.001	89.2	0.501 (FG)	<0.001	92.6	0.478 (FG)	<0.001	99.1	0.248 (P)	<0.001	90.2	0.415 (P)	<0.001
All	7428	72.9	0.701 (FG)	<0.001	92.5	0.807 (E)	<0.001	92.0	0.522 (FG)	<0.001	99.6	0.316 (P)	<0.001	94.0	0.448 (P)	<0.001

* Insufficient scoring variation to calculate kappa coefficients (all indicator scores were 0). Kappa interpretation: ≥0.75 ‘excellent’ (E), 0.40–0.74 ‘fair to good’ (FG), and <0.40 ‘poor’ agreement (P) [10].

##### Breathing

Rabbits showing signs of breathing during bleeding were observed in all SHs. The highest prevalence of breathing was found in SH-3 (289/398; 72.6%) and the lowest in SH-15 (25/400; 4.6%) as shown in Table 6. The observers differed significantly in the detected prevalence of breathing in 12 out of the 16 SHs evaluated (*p* < 0.001). This is mainly because Obs-D had a lower angle of visibility, preventing the proper observation of both the rabbits’ flanks and mouth movements to detect breathing since there was not enough space available to assess the rabbits in a dorsolateral or ventrolateral position in the SHs. Therefore, these observers had to assess the rabbits on either dorsal or ventral position according to the SH. The PoA between observers ranged from 52.3 to 97.8% whereas κ ranged from 0.43 to 0.83 and was interpreted in 12 SHs as “fair to good” and in four SHs as “excellent” agreement (Table 7).

Considering the data from all SHs, the detection of breathing differed statistically among evaluators (*p* < 0.001; Table 6). However, by calculating the average of the three evaluators who provided similar scores, a general prevalence of 28.4% was obtained. Nonetheless, the PoA was 92.5% and the κ was statistically significant and interpreted as being in “excellent” agreement with the observers (*p* < 0.001; κ = 0.85; Table 7).

##### Spontaneous Blinking

Rabbits with spontaneous blinking during bleeding were observed in all SHs. However, the observers differed significantly in the detected prevalence in 11 out of the 16 SHs evaluated (*p* < 0.001). Similarly to the detection of breathing, Obs-D detected significantly fewer animals with spontaneous blinking due to a less favourable position for observing the rabbits’ eyes than other observers. The highest prevalence of this sign of consciousness was found in SH-3 (53/400; 13.3%) and the lowest in SH-5 (3/308; 0.8%) as shown in Table 6. The PoA between observers ranged from 75.8 to 99.2%, whereas κ ranged from 0.28 to 0.67 and was interpreted as “poor” in four SHs and as “fair to good” in 12 (Table 7).

When considering all the rabbits assessed during bleeding, the detection of spontaneous blinking differed statistically between observers (*p* < 0.001; Table 6) but the average of the closest outcomes revealed a prevalence of 5.7%. On the other hand, the PoA was 92.0% and the κ was statistically significant and interpreted as “fair to good” among observers (*p* < 0.001; κ = 0.52; Table 7).

##### Vocalisation

Vocalisations were heard in rabbits from 8 out of the 16 SHs evaluated. The observers did not differ significantly in the detected prevalence (*p* > 0.05) but, in one SH (i.e., SH-15), Obs-C heard three rabbits vocalising but the other Obs did not hear any (Table 6). The highest prevalence of this sign of consciousness was found in SH-6 (2/294; 0.7%), as shown in Table 6. The PoA between observers ranged from 99.1 to 100% and κ could not be computed in eight SHs due to absence of detected vocalisations. In the remaining eight SHs assessed, the κ ranged from 0.00 to 0.67 and was interpreted in six SHs as “poor” and in two SHs as “fair to good” agreement (Table 7).

In all the rabbits assessed during bleeding, the prevalence of rabbits vocalising was similar between observers (*p* > 0.05; Table 6) and the mean prevalence was 0.13%. The PoA was 99.6% and the κ was statistically significant and interpreted as “poor” between observers (*p* < 0.001; κ = 0.32; Table 7).

##### Righting Reflex

Rabbits showing righting reflex were observed in 14 out of the 16 SHs evaluated. The observers differed significantly in the detected prevalence in 12 SHs (*p* > 0.05). The highest prevalence of this sign of consciousness was found in SH-5 (46/380; 12.1%), as shown in Table 6. The PoA between observers ranged from 86.7 to 100% and κ could not be computed in one SH due to the absence of detected righting reflexes. In the remaining 15 SHs assessed, the κ ranged from 0.00 to 0.66 and was interpreted in eight SHs as “poor” and in seven SHs as “fair to good” agreement (Table 7).

As a general result, the prevalence of rabbits with righting reflex differed statistically between observers (*p* < 0.001; Table 6) but the average of the closest outcomes revealed a prevalence of 3.3%. The PoA was 94.0% and the κ was statistically significant and interpreted as “poor” agreement between observers (*p* < 0.001; κ = 0.45; Table 7).

### 3.2. Relationship Among Animal-Based Indicators

#### 3.2.1. Immediately After Stunning

The proportions of rabbits exhibiting outcomes of consciousness and the combinations of ABIs for the same rabbit are illustrated in a Venn diagram (Figure 4).

The absence of tonic–clonic seizure was the most frequently observed indicator, followed by presence of breathing and spontaneous blinking. Vocalisation was absent in all rabbits assessed. Combinations of more than one sign of consciousness included the absence of tonic–clonic seizure along with either breathing or spontaneous blinking (Figure 4A). No rabbit simultaneously displayed the three outcomes of consciousness.

#### 3.2.2. During Bleeding 

The proportions of rabbits showing outcomes of consciousness and their combinations at individual level is shown as a Venn diagram in Figure 4B. This diagram showed that signs of breathing was the most frequent evidence of consciousness observed, followed by spontaneous blinking, righting reflex, and, to a lesser extent, vocalisations. Furthermore, some rabbits simultaneously showed two of the four indicators of consciousness evaluated. The most frequent combinations were breathing and spontaneous blinking, and breathing and righting reflex. Other combinations found, but less observed, included spontaneous blinking and righting reflex, and breathing and vocalisations. On the other hand, some rabbits were also observed with three of the four indicators of consciousness assessed simultaneously, the combination observed being breathing, spontaneous blinking, and righting reflex.

### 3.3. Relationship Between Key Parameters and Stunning Efficiency

The risk factors related to management conditions at slaughter known to be linked to ineffective stunning were analysed. These factors included not wetting the rabbits’ heads prior to stunning, the number of stunners, and long stun-to-stick intervals. Additionally, combinations of electrical parameters applied to batches of varying characteristics, such as genetics and average body weight of the batch, were examined. The mean prevalence and 95% confidence interval of the closest outcomes between observers regarding the failure to induce unconsciousness in rabbits (i.e., observations made immediately after stunning) and the prevalence of rabbits regaining consciousness during bleeding (i.e., observations made during bleeding) were calculated. In addition, the management procedures and key stunning parameters per SH and batch are shown in Table 8.

#### 3.3.1. Immediately After Stunning

All rabbits were effectively stunned in SH-6, SH-7, SH-8, SH-11, SH-13, and SH-15 (in all batches assessed), as no rabbit showed signs of consciousness (Table 8). Meanwhile, SH-5 showed the highest prevalence of failure at inducing unconsciousness with 15.3% ([8.9–18.4] 95% CI) of rabbits showing at least one ABI in batch 1. The layout of SH-3 and the design of the stunner that automatically bled the rabbits in SH-10 did not allow the assessment of the state of consciousness immediately after stunning and before bleeding and, therefore, no information could be provided on these variables.

#### 3.3.2. During Bleeding

In all the 16 SHs, each batch had some rabbits with at least one sign of consciousness. The prevalence of rabbits showing outcomes of consciousness varied greatly between batches and SHs. The lowest prevalence during bleeding was observed in SH-10 (batch 1: 2.3% [0.6–7.9%] 95% CI; batch 2: 2.6% [1.5–4.4%] 95% CI), while the highest was observed in SH-3 (batch 1: 92.9% [77.4–98.0%] 95% CI; batch 2: 71.4% [66.5–75.7%] 95% CI) as shown in Table 8.

### 3.4. Factors That Influence the Stunning Efficiency

Multicollinearity was negligible among the potential factors influencing efficient stunning as all VIFs were below 1.16.

The factors influencing the efficiency of HOES in rabbits are shown in Table 9. Specifically, a stun-to-stick interval below 5 s had the largest effect, reducing the odds by 92% (OR = 0.08), followed by wetting the rabbit’s head with a 34% reduction (OR = 0.66). The combination of high current and low frequency (>200 mA and 50 Hz) and high current and high frequency electrical parameters (>200 mA and 50 < x ≤ 300 Hz) showed reductions of 58% (OR = 0.42) and 39% (OR = 0.61), respectively. The intercept, with an OR of 1.81, indicates that, when the stun-to-stick interval is longer than 5 s, the lateral cut is performed, the heads of the rabbits are not wet when stunning, and the electrical parameters are considered suboptimal (i.e., <200 mA and >50 Hz), the odds of inefficient stunning are 1.81 times higher compared to when none of these risk factors are present.

## 4. Discussion

One of the objectives of this study was to investigate the IOR of valid and feasible ABIs to assess the state of consciousness following HOES in rabbits. This study also aimed to report the prevalence of indicators, as well as the failure to induce and maintain unconsciousness in commercial slaughterhouses, and to identify factors that influence stunning efficiency.

This study compared the assessment of four observers on 11,540 rabbits across 38 batches from 16 different SHs and different key stunning parameters applied from the three main rabbit producer countries in the EU-27 [1]. Even though SHs were not randomly sampled per se, they represent quite a variety in terms of slaughter capacities, equipment designs, key electrical parameters combinations, stun-to-stick intervals, and line speed. Furthermore, these 16 SHs alone slaughter most of the rabbits reared in the EU.

Regarding the observers, all were co-authors of the present study, highly trained, and had prior experience conducting similar research on broiler chickens [17] and turkeys [18]. Additionally, they reached a consensus on the definition of the ABIs before conducting the assessments. The number of observers was kept as high as possible to minimise disruption to the operators and to each other, while ensuring clear visibility for the assessment of the ABIs. To achieve this, the observers were positioned side by side, assessing the same rabbits over the same span of time. However, in some SHs, one or two observers faced difficulties in assessing the state of consciousness due to limited space, which hindered optimal positioning and visibility.

### 4.1. Inter-Observer Repeatability of the Animal-Based Indicators

The inter-observer repeatability of the ABIs was analysed for each individual assessed using both the PoA and κ. High PoA may suggest strong agreement among observers, but this can occur if the sign of consciousness is rarely or hardly ever observed (e.g., vocalisations), leading to agreement when nothing is perceived. In contrast, agreement tends to be lower for signs of consciousness that are more frequently observed (e.g., absence of tonic-clonic seizure and presence of breathing). The interpretation of κ varied slightly depending on the SH assessed for most of the indicators. This happened because κ is strongly influenced by the prevalence of rabbits showing signs of consciousness. The lower the prevalence, the lower the κ may be. However, sometimes, when a sign of consciousness is not detected in any rabbit within a sample, the κ cannot be even computed. These findings suggest that PoA alone does not provide much meaningful information in such cases, and the same applies to κ. However, the combination of PoA and κ provides a more comprehensive view of the performance of an ABI in terms of IOR.

Immediately after stunning, the most repeatable indicators are tonic–clonic seizure, followed by breathing and spontaneous blinking while vocalisation was highly repeatable artificially since it was never heard in any SH.

During bleeding, the most repeatable ABI was breathing, followed by tonic–clonic seizure and spontaneous blinking, while righting reflex and vocalisations were the least repeatable. Despite being the most repeatable, breathing had three main sources of variation in scoring. First, there was hesitation at considering signs of breathing in some SHs where rabbits showed very shallow depth of flank movements accompanied or not by rapid muzzle movements. Second, evaluating breathing was more challenging in batches of dark-furred rabbits compared to white-furred rabbits. Third, sings of breathing were considered when a minimum of two thoracic or abdominal muscle movements associated with breathing were observed. Some rabbits performed the second thoracic or abdominal muscle movements at the end of the observation span, raising doubts on whether it occurred just before or after the established time limit. This highlights the importance of determining an optimal observation period during which more accurate signs of consciousness can be observed in a slaughterhouse. The best position for assessing breathing is ventrolateral, as dorsal or lateral positions may underestimate the prevalence of breathing in rabbits.

It should be noted that the IOR of spontaneous blinking was underestimated in some SHs, affecting the overall PoA and κ values for this indicator. This is because in some SHs, the space available for four observers to have a clear view for detecting spontaneous blinking throughout the assessment period was suboptimal. While two observers could assess spontaneous blinking relatively easily, one or even two of the other observers sometimes reported difficulties in accurately assessing this ABI due to a lack of a proper viewing angle to clearly observe the rabbits’ eyes. Furthermore, although not quantified, it is quite common to find a proportion of rabbits with sealed eyelids due to burns from incorrect electrode placement, potentially underestimating the prevalence of rabbits that would spontaneously blink as a sign of consciousness.

Righting reflex had poor repeatability between assessors, as it can be confounded with preagonal muscle contractions. These episodes are similar to a tonic phase with some muscle tremor and sometimes in combination with the righting of the head. Therefore, the evaluators found it difficult to differentiate this from the righting reflex as voluntary attempts to recover posture.

Vocalisation was the least repeatable indicator of consciousness since it was hardly ever heard and when it seemed to occur there was no consensus among the observers. This was because, when the observers heard vocalisations, they were not able to identify which rabbit it came from and; moreover, this also depended on the hearing ability of the evaluator. In addition, it is important to highlight that the noise level in the SHs is likely to impair the vocalisation assessment. This is because the vocalisations in bled rabbits are not high-pitched or loud and, therefore, not clearly detectable unless the assessor is within a few centimetres of the animal’s head, but this was not the case in the assessments carried out in the present study. In contrast, for rabbits that were stunned but, for whatever reason, escaped from the bleeder, once they regained consciousness, high pitch vocalisations were clearly detected for all at a distance despite the different hearing abilities and the ambient noise of the SH and, therefore, this indicator should not be neglected despite being poorly repeatable at this stage.

Comparing the inter-observer repeatability (IOR) of the common ABIs assessing the state of consciousness in the present study (head-only electrically stunned rabbits) with those reported for water-bath stunned broiler chickens [17] and turkeys [18], slight variations were observed. Before bleeding, the IOR of tonic seizure was higher in turkeys (κ = 0.75) than in rabbits (κ = 0.67) and broilers (κ = 0.64). A similar trend was observed for breathing, with the highest IOR in turkeys (κ = 0.75), followed by rabbits (κ = 0.63) and broiler chickens (κ = 0.58). However, spontaneous blinking showed greater IOR in rabbits (κ = 0.41) than in turkeys (κ = 0.15) or broilers (κ = 0.14). Vocalisation was not observed in any rabbits, broiler chickens, or turkeys during this key stage. During bleeding, the IOR of breathing was nearly identical for turkeys (κ = 0.82) and rabbits (κ = 0.81), both of which were higher than in broiler chickens (κ = 0.64). The higher repeatability of most indicators can be partly attributed to the lower line speed observed in turkeys (300 to 3000 turkeys/hour) and rabbits (600 to 3600 rabbits/hour) compared to the higher speeds seen in broiler chicken slaughterhouses (200 to 10,500 chickens/hour). Slower line speeds appear to improve the consistency and IOR of assessing the animal’s state of consciousness.

Achieving higher IOR is likely due to comprehensive training and broader testing. Consequently, better training emerges as a key factor in improving animal welfare assessments in SHs. This can be accomplished through a combination of theoretical lectures, instructional videos, and practical field exercises within SHs, enabling trainees to align their scoring with that of experts [19].

### 4.2. Relationship Among Animal-Based Indicators

Effective electrical stunning is known to induce epileptiform activity in the brain which manifests as tonic–clonic seizure, cessation of breathing, and absence of physical and behavioural reflexes [20,21]. In this study, 58 out of 4112 rabbits showed flaccid muscle tonicity instead of tonic–clonic seizure following electrical stunning. According to Anil et al. [3], some of these animals may exhibit normal EEG patterns and are therefore considered conscious, while others may display epileptic EEG patterns, indicating unconsciousness. Therefore, effective induction of unconsciousness is only confirmed by the observation of tonic–clonic seizure immediately after stunning and the presence of flaccidity is not indicative of unconsciousness. Furthermore, 17 animals out of 4112 showed signs of breathing and 8 showed signs of spontaneous blinking during or after the tonic–clonic seizure and before neck cutting. The presence of these indicators suggests a failure to achieve unconsciousness. During bleeding, the presence of tonic–clonic seizure indicates that the rabbit is still unconscious and usually it means that the stun-to-stick interval was short (with the tonic–clonic phase still persisting), reducing the likelihood of regaining consciousness. However, the absence of these seizures does not imply consciousness, as the rabbit may have already experienced them shortly before. A rabbit is considered regaining consciousness only when it shows absence of tonic–clonic seizure along with at least one of these indicators of consciousness: breathing, spontaneous blinking, and vocalisations.

Some indicators of consciousness were observed simultaneously in the same rabbit. The most frequent combinations included breathing with spontaneous blinking and breathing with the righting reflex. A triple combination of breathing, spontaneous blinking, and the righting reflex was also noted. It seems that when a rabbit starts breathing, it is more likely to spontaneously blink or attempt to regain posture (righting reflex) later on. Nevertheless, some rabbits exhibited spontaneously blinking without breathing or righting reflex, indicating that the order of appearance of the signs of consciousness varied. Rabbits displaying breathing, spontaneous blinking, or the righting reflex are considered conscious, indicating their ability to experience pain, fear, and distress during slaughter.

On the other hand, rabbits that are scored as showing a righting reflex but do not breathe or blink spontaneously cannot be considered conscious, and it is more likely to be what is known as Lazarus’ sign described in humans [22] as well as in slaughtered animals [23]. The Lazarus sign, named after the biblical man who rose from the dead, refers to spinal reflexes and automatisms observed in individuals with apparent brain death. These reflexes in humans can include spontaneous head turning or shaking, neck-arm flexion, neck-hip flexion, neck-abdominal flexion, arm extension, and elbow and finger flexion that mimic voluntary grasping or clasping. The Lazarus sign is a reflex mediated by a reflex arc neural pathway which passes via the spinal column but not through the brain. As a consequence, the movement is possible in brain-dead patients. The reflex is often preceded by slight shivering motions. When the rabbit is bleeding out, the lack of oxygen can cause dysfunction in the central nervous system, but spinal reflexes can remain active for a time. They include movements such as muscle contractions or spasms, which can resemble the righting reflex in rabbits. This similarity not only makes the righting reflex a poor inter-observer repeatable but also unreliable as an indicator of consciousness unless accompanied by breathing and/or spontaneous blinking.

### 4.3. Relevant Animal-Based Indicators

Animal-based indicators must be relevant for use in welfare assessments. In this context, relevance means that the indicators should be valid, feasible, and repeatable. In the present study, the ABIs recommended by the EFSA [3] were chosen based on their validity and feasibility; however, their repeatability had not yet been tested. By considering these three factors, along with the validity demonstrated by the relationship between ABIs in this study, we can confidently recommend a set of indicators for each key stage.

Immediately after stunning, the relevant indicators are tonic–clonic seizure, breathing, and spontaneous blinking. The presence of tonic–clonic seizure is crucial to confirm that a rabbit has been effectively stunned and rendered unconscious. Rabbits that showed an absence of tonic seizure failed to be induced to unconsciousness. Rabbits that showed signs of breathing and/or spontaneous blinking and/or vocalisations recovered the state of consciousness before being bled.

During bleeding, the relevant indicators are breathing, spontaneous blinking, and vocalisations. Rabbits that showed signs of breathing and/or spontaneous blinking and/or vocalisations recovered or were in the process of recovering to a state of consciousness after bleeding. The righting reflex should only be considered an indicator of consciousness when it is accompanied by breathing and/or spontaneous blinking to avoid confusing it with the Lazarus sign. Rabbits that perform righting reflex while breathing and/or vocalise have regained a high level of consciousness and are, therefore, fully capable of experiencing pain, distress, and suffering.

### 4.4. Relationship Between Electrical Parameters and Stunning Efficiency

Effective stunning is achieved when a rabbit is rendered unconscious and remains in this state during bleeding until death occurs. To ensure this, the state of consciousness was assessed immediately after stunning and during bleeding.

One of the objectives of the present study was to compare the efficiency of stunning across different SHs and key parameters (i.e., current, voltage, frequency, time of exposure, and stun-to-stick interval) when using a head-only electrical device on rabbits. However, numerous other factors can influence the effectiveness of stunning in rabbits. These factors include animal characteristics (e.g., genetics and body weight), operator management practises (e.g., wetting the rabbits’ heads prior to stunning, regular maintenance, cleaning of the electrical equipment, and staff fatigue) and the type of bleeding (ventral or lateral).

The results showed that, despite the divergence in the applied electrical parameters and exposure time, the induction of unconsciousness was effective in almost all batches evaluated in the different SHs. However, in some batches, the effectiveness of inducing unconsciousness was low (e.g., SH-1 batch 1, SH-4 batch 4, and SH-5 batch 2). According to our observations, poor electrical contact was the main cause of rabbits showing signs of consciousness immediately after stunning. This occurs when the electrodes are worn out or covered with debris. In one of the SHs visited, the fur trapped in the device not only impaired the electrical contact, causing a failure to induce unconsciousness in some rabbits, but also a small spark due to burning the trapped fur. The heads of the animals were not wet in any of the SHs with low stunning efficiency. The rabbits without tonic–clonic seizure and/or signs of breathing and/or spontaneous blinking are at a high risk of experiencing pain, distress, and suffering when shackled, being bled, and during bleeding.

It should be highlighted that some rabbits received pre-stun shocks due to incorrect contact between the rabbit’s head and the stunner electrodes which is very painful for conscious animals. In these animals, muscle tonicity did not occur, and operators repeated contact with the animal and the stunner until tonicity occurred as a sign of unconsciousness. Only then were the rabbits shackled on the SH line.

In this study, the prevalence of rabbits that were not effectively rendered unconscious was recorded from the moment they were shackled. As a result, cases where rabbits were not successfully rendered unconscious on the first attempt and needed to be re-exposed to the electrical stunning device were not specifically identified. Therefore, there is no estimation of the frequency of re-stunning before shackling.

Although unconsciousness was effectively induced in almost all rabbits across all batches (4035 out of 4112, 98.1%), the prevalence of rabbits regaining consciousness varied strongly, ranging from 2.3% to 92.9% depending on the batch and the SH assessed.

We believe that the variability observed is primarily due to differences in SH design, the type of stunning device used, and slaughterhouse-specific practises such as the electrical parameters applied, the maintenance and calibration of the devices, and likely the level of animal welfare training provided to the operators, among others.

The lowest prevalence was found in SH-10 (batch 1: 2.3% [0.6–7.9%]; batch 2: 2.6% [1.5–4.4%]). This was likely because the stunner they used not only wet the rabbit’s head and stunned but also bled immediately after stunning if the operator pressed a button (once the operator noticed the tonic seizure), resulting in an extremely short stun-to-stick interval. The type of cut was ventral, allowing for a more rapid death by exsanguination and reduced the chance of regaining consciousness before death. The stunning device used in this SH was unique and offered by far the best results, as rabbits died so quickly that they exhibited only muscle tonicity without progressing to clonic seizures. However, one drawback of this system is that it does not provide a means to verify the efficiency of stunning (except by the presence of tonic seizure) before bleeding, which could potentially lead to the risk of bleeding conscious rabbits.

After SH-10, the SH with the lowest prevalence of rabbits showing at least one indicator of consciousness in a batch was SH-7. SH-7 was equipped with a unique stunner, different from the most commonly observed ones. It featured an electrified grid mounted on the wall, where the operator would make contact with the rabbit’s head, then hang it on the shackle line, and another operator would bleed the animal. Due to SH-7′s low throughput capacity, there was only one operator stunning and hanging and one for bleeding the rabbits, resulting in a very short stun-to-stick interval (approximately 2 s). The electrical parameters at SH-7 seemed appropriate, providing high currents and low frequencies to ensure a sufficient unconsciousness time for the rabbits to die without regaining consciousness. However, variation was observed in the prevalence of rabbits regaining consciousness between batches (batch 1: 4.3% [2.8–6.6]; batch 2: 14.9% [9.9–21.9]). Although both batches had the same genetics, the differences might be attributed to the variance in average body weight between batches (batch 1: 2.3 kg; batch 2: 1.7 kg). The lighter rabbits received a lower current (batch 1: 610 ± 171 mA/rabbit; batch 2: 498 ± 135 mA/rabbit). Although it cannot be for certain, this might be explained by the fact that the electrical tongs were designed for heavier rabbits (with larger heads), so they did not sufficiently press on the heads of smaller rabbits leading to impaired electrical contact. On the other hand, in this SH, the rabbits were wetted with a pressurised hose controlled by an operator while still in stacked transport containers. Consequently, it is difficult to ensure that all the rabbits had their heads wetted, as it is likely only those positioned closest to the container openings were wetted. In addition, the type of cut was lateral rather than ventral, which could lead to prolonging the time it takes to die and, therefore, a greater chance of the rabbits regaining consciousness before death.

In SH-16, the prevalence in the sample ranged between 7.1 and 15.7% according to the batch assessed. The SH-16 was equipped with three stunners, so three operators were in charge of stunning rabbits while two operators bled the rabbits on the line. The stunners consisted of electrical tongs that were fixed in a channel-shaped support. The rabbits were not wetted before stunning. The stun-to-stick interval varied considerably (from 5 to 20 s) depending on the distance between each stunner and the bleeding operator. In this context, rabbits stunned with the stunner closest to the bleeding operator likely had a lower risk of regaining consciousness compared to those stunned farther away, due to the shorter stun-to-stick interval. Despite the long stun-to-stick interval for most of the rabbits, the prevalence of rabbits regaining consciousness was unexpectedly lower than in other SHs with similar characteristics. This may be attributed to the apparently optimal electrical parameters (high current: >400 mA/rabbit, low frequency: 50 Hz), which likely ensured a prolonged period of unconsciousness. Additionally, the ventral cuts performed are presumed to increase bleeding speed and shorten the time it takes to die, thereby reducing the likelihood of rabbits regaining consciousness before death.

For the rest of the SHs assessed, the samples assessed had a prevalence between 20% and 92.9% of rabbits showing at least one indicator of consciousness (in SH-1, SH-2, SH-3, SH-4, SH-5, SH-6, SH-9, SH-12, SH-13, SH-14, and SH-15).

The reported range of prevalence for broiler chickens regaining consciousness during bleeding within a batch was [0–11%], while for turkeys it was [0–16%], and for rabbits, it ranged from [2–93%]. Animals that regain consciousness are at high risk of experiencing pain and fear. On the one hand, these prevalences reported underscore the critical importance of implementing proper monitoring and intervention protocols. It should be highlighted again that HOES is the most commonly used method in rabbits [2], but it appears to be insufficient in protecting their welfare at the time of slaughter. Consequently, it is crucial to investigate the factors influencing stunning efficiency to minimise welfare risks and implement improvements. Alternatively, alternative stunning methods that offer better welfare outcomes than HOES that could be approved for use in the EU should be considered.

### 4.5. Factors That Influence the Stunning Efficiency

The risk of rabbits showing at least one indicator of consciousness during bleeding is significantly influenced by several factors, with the stun-to-stick interval being the most critical. Intervals exceeding 5 s and not wetting the rabbits’ heads all exacerbate this risk of rabbits showing at least one indicator of consciousness. It is well-established that the higher the current and the lower the frequency applied in electrical stunning, the more efficient the stunning method is in inducing and maintaining unconsciousness [23]. The European Commission [6] cites thresholds of over 140 mA and over 400 mA for rabbits, while the model selected in the present study considers a current below 200 mA as increasing the odds of ineffective stunning. It is important to note that the more risk factors are present in an SH, the higher the likelihood of ineffective stunning in rabbits.

Although not quantified in the present study, other factors that undoubtedly influence the stunning efficiency is the regular cleaning and maintenance of the stunning device.

It is important to note that there is potential for improvement in the system used for wetting the rabbit’s head. This process is not consistently carried out for all rabbits, primarily due to issues with the wetting system itself. Additionally, the maintenance of the wetting system should be carefully monitored, as some slaughterhouses had sprinklers on the conveyor belt that either dripped water or malfunctioned, leading to rabbits not getting wet or being wetted with a hose. It is crucial that only the rabbit’s head is wetted, not its body. Wetting the body can lead to cold stress, significantly compromising the rabbit’s welfare, particularly when this is not done just before stunning or the environmental temperature is low (e.g., during winter).

## 5. Conclusions

Immediately after stunning, the relevant indicators are tonic–clonic seizure, breathing, and spontaneous blinking. The presence of tonic–clonic seizure is crucial to confirm that a rabbit has been effectively stunned and rendered unconscious. Rabbits that do not exhibit tonic–clonic seizure and/or display breathing and/or spontaneous blinking are at high risk of experiencing pain, fear, and distress when shackled, bled, and during the bleeding process.

During bleeding, the relevant indicators are breathing, spontaneous blinking, and vocalisations. The righting reflex should be considered an indicator of consciousness only when it is accompanied by breathing and/or spontaneous blinking. The most observed combinations are breathing and spontaneous blinking, and breathing and righting reflex. Rabbits that display breathing and/or spontaneous blinking during bleeding are also at high risk of experiencing pain, fear, and distress.

Although unconsciousness is effectively induced in nearly all rabbits, indicators of consciousness are frequently observed after neck-cutting, suggesting that a variable but significant proportion of rabbits are progressively recovering consciousness before death in all slaughterhouses.

The key factors that contribute to effective stunning (i.e., the animal is rendered unconscious and does not regain consciousness before death) in rabbits include ensuring the stun-to-stick interval is less than 5 s, using a current above 200 mA with a frequency of 50 Hz, and wetting the rabbits’ heads prior to stunning. While not quantified, the regular cleaning and maintenance of stunning devices also appear to play a crucial role in achieving effective stunning.

The findings of this study have practical implications for refining welfare monitoring protocols and improving slaughter practises for rabbits under commercial conditions. By providing a detailed list of the relevant indicators and identifying the key factors influencing stunning efficiency, this research contributes to the development of standardised protocols aimed at enhancing rabbit welfare at the end of life.

## Figures and Tables

**Figure 1 animals-15-00587-f001:**
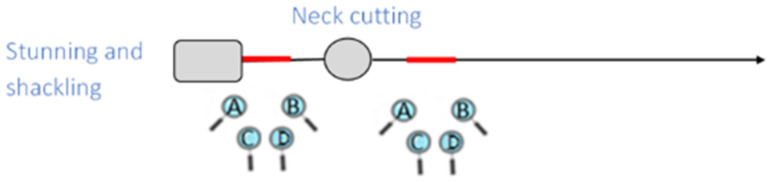
Position of the four observers during the assessment of animal-based indicators of the state of consciousness after head-only stunning in rabbits. The position of the lens represents the position of the observers (i.e., immediately after stunning and during bleeding) and the red segments are the observation area.

**Figure 4 animals-15-00587-f004:**
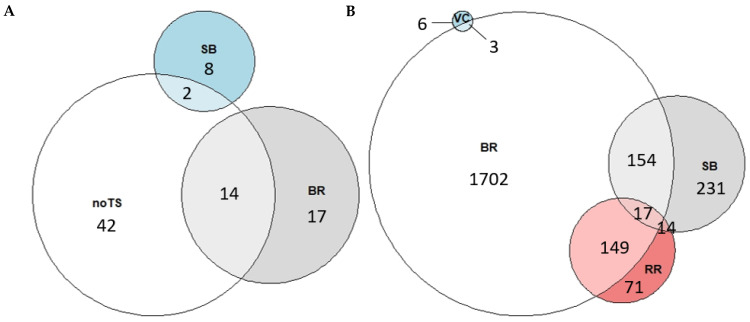
Venn diagram of the outcomes of consciousness observed in the animal-based indicator assessed after head-only electrically stunned rabbits (**A**) immediately after stunning and (**B**) during bleeding. The outcomes of consciousness are as follows: No TS: the absence of tonic–clonic seizure; BR: presence of breathing; SB: presence of spontaneous blinking; VC: presence of vocalisation; and RR: presence of righting reflex. The numbers specify the total amount of rabbits showing each sign of consciousness or combinations of outcomes of consciousness from a total of 4112 rabbits assessed immediately after stunning and 7428 during bleeding.

**Table 1 animals-15-00587-t001:** Main characteristics of the sixteen rabbit slaughterhouses (SH) included in this study.

SH	Line Speed, Rabbits/h	Wetting Heads Before Stunning, Yes/No	Stunners in Use Simultaneously, n	Stun-to-Stick Interval According to the Stunner/s, s	Bleeding Method *	Bleeding Cut	Number Operators Bleeding, n
1	800	No	2	10 and NA	M	Unilateral	1
2	1500	NA	2	11 and NA	M	NA	2
3	1600	No	1	22	M	Unilateral	1
4	2600	No	3	15, 10 and 8	M	Unilateral	1
5	2100	No	4	36, 30, 24 and 19	M	Bilateral	2
6	700	No	1	15	M	Bilateral	1
7	700	Yes	1	2	M	Unilateral	1
8	600	No	1	3	M	Unilateral	1
9	1850	Yes	3	18, 12 and 7	M	Bilateral	1
10	1400	Yes	3	0	A	Bilateral	1
11	700	NA	1	3	M	Bilateral	1
12	800	Yes	1	16	M	Unilateral	1
13	1700	No	3	25, 19 and 6	M	Bilateral	1
14	1920	Yes	3	33, 24 and 17	M	NA	1
15	3200	Yes	4	22, 20, 18 and 13	M	Unilateral	2
16	3600	No	3	20, 13 and 5	M	Bilateral	2

* Bleeding method: M (manually); A (automatically); SH: slaughterhouse; NA: data not available.

**Table 2 animals-15-00587-t002:** Number of batches slaughtered, category, and average body weight of the rabbits in the batches evaluated for each slaughterhouse. The average electrical parameters for each of the head-only electric stunners (±standard deviation) and the exposure time of the electric tongs on the head of the rabbits (±standard deviation) are also reported.

		Characteristics of the Rabbits				Stunning Parameters Used			
SH	Batch	Animal Category	BW, kg/Rabbit	No Rabbits	Stunner	Current, mA/Rabbit	Frequency, Hz	Voltage, V	Time, ms
1	1	M	3.00	1200	1	300 ± 83	401	396	2440 ± 24
					2	284 ± 88	401	394	2340 ± 34
	2	M	2.75	2000	1	282 ± 78	401	397	2430 ± 30
					2	280 ± 84	401	397	2360 ± 26
2	1	M	2.45	5150	1	866 ± 122	50	182	1030 ± 112
					2	843 ± 125	50	183	921 ± 120
	2	M	2.60	615	1	893 ± 118	50	179	1042 ± 138
					2	818 ± 121	50	182	903 ± 120
	3	M	NA	2050	1	812 ± 107	50	181	920 ± 121
					2	864 ± 109	50	181	915 ± 113
3	1	M	2.4	4022	1	675 ± 94	50	277	1231 ± 161
					2	1059 ± 275	50	265	984 ± 138
	2	M	2.3	10,610	1	787 ± 241	50	277	1286 ± 597
					2	1044 ± 286	50	267	1014 ± 152
	3	B	4.4	120	1	753 ± 211	50	280	3473 ± 539
4	1	M	2.8	4739	1	779 ± 229	50	302	732 ± 111
					2	795 ± 204	50	311	642 ± 76
					3	1006 ± 268	50	293	877 ± 105
		B	4.4	276	1	1033 ± 267	50	299	6770 ± 292
					2	955 ± 205	50	310	7780 ± 236
					3	1189 ± 348	50	290	8075 ± 315
	2	M	2.3	1475	1	648 ± 202	50	304	683 ± 110
					2	926 ± 299	50	309	657 ± 64
					3	960 ± 244	50	292	863 ± 90
	3	M	2.5	6107	1	1100 ± 252	50	297	808 ± 69
					2	922 ± 262	50	308	824 ± 165
					3	797 ± 249	50	294	772 ± 99
	4	M	2.4	156	1	1040 ± 259	50	289	788 ± 106
					2	764 ± 212	50	295	648 ± 112
					3	695 ± 204	50	292	725 ± 85
5	1	M	2.4	5273	1	792	NA	350	1396
					2	713	NA	349	1324
					3	577	NA	350	1307
					4	585	NA	356	1326
	2	M	2.6	5416	1	670	NA	353	1311
					2	628	NA	352	1229
					3	576	NA	350	1369
					4	433	NA	359	1180
6	1	M	2.1	1819	1	438 ± 128	50	180	496 ± 253
	2	M	2.1	576	1	445 ± 191	50	180	503 ± 263
7	1	M	2.3	2160	1	610 ± 70	50	145	567 ± 121
	2	M	1.7	443	1	498 ± 135	50	142	485 ± 107
8	1	M	2.1	1488	1	467 ± 175	50	210	509 ± 152
9	1	M	2.0	1401	1	742 ± 243	50	244	923 ± 307
					2	679 ± 264	50	247	935 ± 263
					3	723 ± 274	50	258	783 ± 367
	2	M	1.9	1736	1	764 ± 249	50	244	998 ± 268
					2	606 ± 261	50	249	907 ± 238
					3	742 ± 244	50	258	708 ± 254
	3	M	ND	3080	1	742 ± 234	50	243	938 ± 219
					2	630 ± 257	50	247	780 ± 222
					3	681 ± 231	50	258	634 ± 193
10	1	M	2.3	5300	1	434 ± 13	300	145	701
					2	404 ± 100	300	158	701
	2	M	2.6	2790	1	407 ± 97	300	156	701
					2	405 ± 95	300	179	701
11	1	M	2.4	1320	1	138 ± 17	150	95	400
12	1	M	2.8	1800	1	196 ± 43	150	112	274 ± 31
		B	4.7	50		208 ± 38	150	116	402 ± 17
	2	M	2.7	1904	1	183 ± 57	150	127	386 ± 46
		B	4.6	66		198 ± 57	150	139	388 ± 64
13	1	M	2.6	4800	1	127 ± 20	250	108	700
					2	127 ± 27	250	110	700
					3	124 ± 31	250	103	700
	2	M	2.9	3131	1	141 ± 18	250	86	700
					2	132 ± 22	250	99	700
					3	128 ± 23	250	102	700
		B	4.0	63	2	143 ± 13	250	79	700
					3	143 ± 8	250	81	700
14	1	M	3.0	1035	1	172 ± 37	250	152	300
					2	166 ± 41	250	170	300
					3	170 ± 37	250	163	300
	2	M	2.6	2610	1	174 ± 34	250	154	300
					2	173 ± 42	250	164	300
					3	173 ± 38	250	161	300
		B	4.7	120	1	173 ± 36	250	153	300
					2	168 ± 41	250	169	300
					3	174 ± 42	250	164	300
	3	M	3.0	3390	1	173 ± 31	250	154	300
					2	183 ± 40	250	153	300
					3	174 ± 44	250	156	300
15	1	M	2.8	5150	1	256 ± 68	50	122	384 ± 97
					2	242 ± 65	50	116	367 ± 88
					3	279 ± 65	50	116	554 ± 99
					4	250 ± 56	50	110	369 ± 72
	2	M	2.4	5488	1	274 ± 70	50	123	437 ± 132
					2	273 ± 76	50	116	450 ± 127
					3	248 ± 69	50	117	439 ± 13
					4	252 ± 53	50	110	520 ± 10
16	1	M	2.3	696	1	499 ± 122	50	210	437 ± 10
					2	641 ± 146	50	208	503 ± 50
					3	670 ± 198	50	204	406 ± 10
	2	M	2.0	7100	1	535 ± 157	50	211	469 ± 98
					2	518 ± 141	50	213	453 ± 64
					3	631 ± 156	50	204	396 ± 96
		B	4.2	40	1	503 ± 142	50	211	411 ± 10
					2	613 ± 154	50	208	494 ± 63
					3	624 ± 200	50	204	376 ± 100
	3	M	2.2	3200	1	517 ± 120	50	211	436 ± 67
					2	491 ± 163	50	212	445 ± 117
					3	610 ± 139	50	203	372 ± 73
	4	M	2.3	800	1	486 ± 147	50	212	444 ± 97
					2	549 ± 153	50	212	476 ± 88
					3	601 ± 144	50	204	413 ± 83
	5	M	2.0	8320	1	484 ± 132	50	211	423 ± 64
					2	530 ± 131	50	212	472 ± 83
					3	514 ± 121	50	207	329 ± 61

SH: slaughterhouse; n: number of rabbits in the batch; M: meat; B: breeders; BW: body weight; NA: no data available.

**Table 8 animals-15-00587-t008:** Mean prevalence of rabbits with at least one sign of consciousness immediately after head-only electrical stunning (HOES) and during bleeding and 95% confidence interval (CI) according to the number of rabbits assessed per batch (n) and slaughterhouse (SH: 1 to 16).

					HOES Parameters				Immediately After Stunning			During Bleeding		
SH	Wet,	Stunners, n	Stun-to-Stick Interval, s [min, max]	Batch	Current, mA	Frequency	Voltage	Time	Rabbits	Mean	95% CI	Rabbits	Mean	95% CI
	Yes/No					Hz	V	ms	n	% ¥		n	% ¥	
1	No	2	[NA, 10]	1	291 ± 86	401	395 ± 31	2380 ± 306	193	13.0	[8.9–18.4]	51	51.0	[37.7–64.1]
1	No	2	[NA, 10]	2	281 ± 81	401	397 ± 0	2391 ± 283	298	0.7	[0.2–2.3]	115	37.4	[29.1–46.5]
2	NA	2	[NA, 11]	1	855 ± 124	50	182 ± 3	978 ± 128	200	0.5	[0.1–2.8]	197	20.3	[15.3–26.5]
2	NA	2	[NA, 11]	2	854 ± 125	50	181 ± 3	972 ± 147	33	0.0	[0.0–10.4]	50	16.0	[8.3–28.5]
2	NA	2	[NA, 11]	3	836 ± 111	50	181 ± 3	918 ± 117	100	1.0	[0.2–5.5]	189	23.8	[18.3–30.4]
3	No	1	22	1	875 ± 284	50	270 ± 7	1102 ± 194	0	-	-	28	92.9	[77.4–98.0]
3	No	1	22	2	913 ± 294	50	272 ± 7	1153 ± 460	0	-	-	370	71.4	[66.5–75.7]
4	No	3	[8, 15]	1	860 ± 257	50	302 ± 8	750 ± 138	0	-	-	325	61.2	[55.8–66.4]
4	No	3	[8, 15]	2	858 ± 287	50	301 ± 8	739 ± 129	0	-	-	16	37.5	[18.5–61.4]
4	No	3	[8, 15]	3	939 ± 283	50	299 ± 7	801 ± 120	71	0.0	[0.0–5.1]	0	-	-
4	No	3	[8, 15]	4	834 ± 271	50	292 ± 10	721 ± 117	128	10.2	[6.0–16.6]	292	54.5	[48.7–60.1]
5	No	4	[19, 36]	1	667	NA	351	1338	6	0.0	[0.0–39.0]	192	51.0	[44.0–58.0]
5	No	4	[19, 36]	2	577	NA	354	1272	177	15.3	[10.7–21.3]	188	55.3	[48.2–62.3]
6	No	1	15	1	437 ± 128	50	180 ± 0	495 ± 258	148	0.0	[0.0–2.5]	140	25.7	[19.2–33.5]
6	No	1	15	2	447 ± 135	50	180 ± 0	503 ± 261	128	0.0	[0.0–2.9]	154	29.9	[23.2–37.5]
7	NA	2	2	1	610 ± 171	50	145 ± 4	567 ± 121	381	0.0	[0.0–0.1]	440	4.3	[2.8–6.6]
7	NA	2	2	2	498 ± 135	50	143 ± 4	485 ± 107	0	-	-	134	14.9	[9.9–21.9]
8	No	1	3	1	467 ± 175	50	210 ± 4	509 ± 152	200	0.0	[0.0–1.9]	375	7.2	[5.0–10.3]
9	Yes	3	[7, 18]	1	723 ± 254	50	936 ± 316	250 ± 8	0	-	-	41	39.0	[25.7–54.3]
9	Yes	3	[7, 18]	2	708 ± 258	50	843 ± 270	251 ± 8	12	0.0	[0.0–24.3]	156	51.9	[44.1–59.6]
9	Yes	3	[7, 18]	3	696 ± 244	50	739 ± 239	251 ± 8	142	1.4	[0.4–5.0]	198	35.9	[29.5–42.7]
10	Yes	3	0	1	411 ± 90	300	156 ± 69	669 ± 136	0	-	-	88	2.3	[0.6–7.9]
10	Yes	3	0	2	406 ± 96	300	166 ± 87	665 ± 141	0	-	-	500	2.6	[1.5–4.4]
11	NA	1	3	1	138 ± 17	150	95 ± 34	NA	200	0.0	[0.0–2.8]	548	18.1	[15.1–21.5]
12	Yes	1	16	1	196 ± 43	150	113 ± 56	399 ± 31	200	0.5	[0.0–2.8]	147	43.5	[35.8–51.6]
12	Yes	1	16	2	184 ± 57	150	127 ± 74	386 ± 47	197	0.0	[0.0–1.9]	314	40.1	[34.9–45.6]
13	No	3	[6, 25]	1	126 ± 26	250	107 ± 38	700 ± 0	200	0.0	[0.0–2.8]	122	43.4	[35.0–52.3]
13	No	3	[6, 25]	2	133 ± 22	250	97 ± 33	700 ± 0	198	0.0	[0.0–1.9]	433	38.6	[34.1–43.2]
14	Yes	3	[17, 33]	1	170 ± 38	250	162 ± 34	NA	200	0.5	[0.0–2.8]	5	100	[56.1–100]
14	Yes	3	[17, 33]	2	173 ± 39	250	159 ± 35	NA	0	-	-	386	54.9	[49.9–59.8]
14	Yes	3	[17, 33]	3	177 ± 39	250	155 ± 36	NA	200	0.0	[0.0–2.8]	184	60.9	[53.7–67.6]
15	Yes	4	[13, 22]	1	257 ± 65	50	116 ± 5	419 ± 119	200	0.0	[0.0–2.8]	200	25.5	[20.0–32.0]
15	Yes	4	[13, 22]	2	262 ± 69	50	117 ± 5	461 ± 131	0	-	-	200	21.5	[16.4–27.7]
16	No	3	[5, 20]	1	609 ± 176	50	207 ± 4	449 ± 99	21	0.0	[0.0–15.5]	0	-	-
16	No	3	[5, 20]	2	566 ± 161	50	209 ± 6	437 ± 94	279	0.4	[0.0–2.0]	319	15.7	[12.1–20.1]
16	No	3	[5, 20]	3	529 ± 150	50	209 ± 5	424 ± 95	0	-	-	197	12.7	[8.7–18.1]
16	No	3	[5, 20]	4	557 ± 154	50	209 ± 5	444 ± 92	0	-	-	36	16.7	[7.9–31.9]
16	No	3	[5, 20]	5	508 ± 131	50	211 ± 3	424 ± 88	0	-	-	98	7.1	[2.8–12.7]

¥: mean prevalence of the closest evaluations between four observers; NA: not available.

**Table 9 animals-15-00587-t009:** Factors influencing the efficiency of head-only electrical stunning in rabbits.

Predictors	Odds Ratios	95% Confidence Interval	*p*-Value
(Intercept)	1.81	1.53–2.15	<0.001
Stun-to-stick interval < 5 s	0.08	0.11–0.30	<0.001
Wetting the rabbit’s head	0.66	0.57–0.77	<0.001
Electrical parameters			
>200 mA and 50 Hz	0.42	0.30–0.58	<0.001
>200 mA and >50 Hz	0.61	0.52–0.71	<0.001

## Data Availability

Data are contained within the article.

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
