# Peer review of "Relevant Indicators of Consciousness After Head-Only Electrical Stunning in Rabbits, Stunning Efficiency, and Risk Factors in Commercial Conditions"

_animals, 2025, doi:10.3390/ani15040587_

Round 1
Reviewer 1 Report
Comments and Suggestions for Authors
The study titled "Relevant Indicators of Consciousness after Head-only Electrical Stunning in Rabbits: Stunning Efficiency and Risk Factors in Commercial Conditions" aims to identify the most reliable and animal-based indicators for monitoring consciousness in rabbits following head-only electrical stunning. The topic of the article is highly significant, particularly for its critical implications on animal welfare during the delicate stunning phase. Moreover, the subject matter aligns well with the scope of the journal. Below are my comments for the authors:
General Comments:
I recommend the authors revise the Simple Summary section to align more closely with the journal’s instructions for authors. It should explicitly outline the problem addressed, the aims and objectives, the key findings, the conclusions, and the societal implications of these findings. Additionally, the section should be written in an accessible style for a general audience. I also suggest avoiding the use of abbreviations in this section.
Introduction
The background on EFSA guidelines and stunning methods is well-explained. However, the gap in standardizing indicators of consciousness and their repeatability needs more emphasis to set the stage for the study.
Explicitly mention the lack of robust repeatability data as a critical limitation that the study addresses.
The stated objectives are clear, but the link between assessing indicators and improving slaughterhouse practices could be strengthened.
Add a brief statement on how findings will directly aid in the implementation of standardized monitoring protocols.
Line 14: Abbreviations should be spelled out in full the first time they appear in the text.
Line 51: The citation European Union, 2009 is not included in the references section. Please add it.
Lines 66 and 73: The citation EFSA, 2020 is missing from the references section.
Line 73: The citation Hindle et al., 2010 is also not present in the references section.
Line 100: The abbreviation SH has not been previously introduced. Spell it out in full the first time it is used.
Lines 100–101: I suggest including the number of slaughterhouses per country to provide a clearer context.
While the focus of the study is on rabbit welfare at slaughter, providing a brief overview of welfare concerns during the life and production stages would make the introduction more comprehensive and contextualize the importance of humane stunning practices.
Add a few sentences highlighting key welfare issues in rabbits during rearing, transport, and handling (e.g., housing systems, stress during transport) to emphasize how welfare at slaughter is part of a broader continuum of animal care.
See and cite: 10.3390/ani14203021; doi.org/10.3390/ani14162367 and 10.1080/1828051X.2020.1827990.
Materials and Methods
The authors note difficulties in observer positioning in certain slaughterhouses. This limitation affects the reliability of some ABIs (e.g., breathing and spontaneous blinking).
The Materials and Methods section could benefit from improved clarity and structure to enhance readability.
Provide a clearer explanation of how observer positioning was mitigated statistically or suggest recommendations for future studies.
The large sample size is a strength of the study. However, clarify why SH-3 and SH-10 were excluded from certain analyses due to design limitations.
Add a brief discussion on how the exclusion affects overall conclusions.
The study mentions EFSA-recommended ABIs but does not justify their selection in this context.
Include a rationale for focusing on tonic-clonic seizures, breathing, spontaneous blinking, and vocalization.
The use of Fleiss' Kappa and PoA is appropriate. However, explain why p-values were not adjusted for multiple comparisons when assessing observer variability.
Add a note on controlling Type I errors in multiple comparisons.
Line 286: Please include a relevant citation here, such as the paper 10.3389/fvets.2023.1141286.
Results
The results provide detailed assessments of inter-observer agreement but lack a clear summary of the most repeatable indicators.
Include a short paragraph summarizing which indicators are most reliable and why (e.g., tonic-clonic seizure and breathing).
The high variability in the righting reflex and vocalization is noted. More discussion is needed on the implications of poor repeatability.
Clearly state whether indicators with poor repeatability (e.g., righting reflex) should still be included in practical monitoring protocols.
The large range (2–93%) in rabbits showing consciousness indicators during bleeding requires further contextualization.
Discuss possible slaughterhouse-specific practices (e.g., training, equipment maintenance) that may explain such variability.
The findings related to stun-to-stick intervals and electrical parameters are valuable. However, results for "wetting rabbits' heads" need further discussion on feasibility in commercial conditions.
Assess whether wetting rabbits' heads can be standardized across slaughterhouses.
Tables 2, 6, and 8 should be revised to improve their readability and visual presentation.
Discussion
While the study identifies critical factors influencing stunning efficiency, more emphasis on actionable recommendations for slaughterhouses is needed.
Provide a clear list of recommendations (e.g., maintaining stun-to-stick intervals below 5 seconds, using higher currents, ensuring proper electrode maintenance).
The challenges in inter-observer variability highlight the need for better training.
Propose a standardized protocol for training observers and assessing ABIs in slaughterhouses.
The authors mention similar studies in chickens and turkeys but do not provide comparisons of ABI repeatability across species.
Briefly discuss whether ABIs observed in rabbits align with findings in other species.
The implications of rabbits regaining consciousness before death should be discussed more thoroughly in the context of regulatory compliance and ethical concerns.
Strengthen the argument for stricter monitoring and intervention protocols.
Consider expanding the discussion to include the limitations of the study and its practical applications, which would provide a more comprehensive perspective.
Line 618: The citation Velarde et al., 2002 is not included in the references section and should be added.
Conclusion
The conclusion effectively summarizes the findings but could be more concise.
Highlight the most important actionable points, such as key risk factors and reliable ABIs.
The study highlights several limitations (e.g., observer positioning, exclusion of SH-3 and SH-10). Future research directions should be proposed.
Recommend studies focusing on automation or technology-based ABI assessments to reduce observer bias.
Several cited articles are missing from the reference section and should be added for completeness.
Author Response
Thanks for your valuable feedback and suggestions!

Reviewer 2 Report
Comments and Suggestions for Authors
I want to thank the authors for taking a bold step and delving into animal consciousness and indicators, a really important area. I have some concerns: as the research involved animals, was there anywhere ethical commitee approval obtained for the research?
The simple summary did not report some of the research's findings i.e whether the stated objectives have been achieved. It will be informative if this can be addressed.
L 31-32: statement unclear. Consists of, or were found to consist of?
L 35: statement "sometimes" unclear - at what times were more than one indicators of consciousness observed in a rabbit? what causes this? improper stunning?
L-38-39: Why?
L39 - 45: Unclear, consider revising
L45: First time mentioning the abbreviation SH. Write in full and then follow with the abbreviation please
L82 - 83: Why?
L104: Recommend?
L106: Suggested? Also have these EFSA indicators been used widely from this studies? consider mentioning how impactful they have been used within the literature.
Also, sensitivity for whom? the observers or the animals.
L 107 - 109: How did the authors draw these conclusions? There seems to not be enough literature cited to have drawn these conclusions.
L113 -114: Kindly separate the literature from your research aim. Also, why rabbit? why not other farm animals? what makes rabbit consciousness an interesting topic to study?
L-115 - L119: How does achieving these objectives contribute to the ongoing welfare of rabbits, particularly during stunning? who are the intended audience for this research? what does the refined list aim to do and who does it aim to inform? academics, inspection/enforcement officers or the slaughterhouse workers
L140 - 145: Purposeful and convenient sampling?
Check headings and subheadings numbering.., from 2.3 then 2.3.1 and there is a jump to 3.3.2
L191 - 194: Any backing for this in the literature?
L195 - 204: Consider creating "limitations of the study" as some information here can go under that.
Also does it mean that a rabbit could have been observed more than once if missed by a certain observer? or do these fall under those removed from the analysis?
L207 - 213: Are these the only indicators? why did you chose these? are they the most widespread in the literature or are you sticking with them based on EFSA' study?
L213- 215: repeatability is not consistently defined and useddd. at one point it seems repeatability is that of the indicators themselves. But then, here, the discourse turns to inter-observer repeatability.
The trained observers... are they part of the co-authors? or were they separate people? If separate, did the researcher choose the list and handed over the list of indicators for them to agree on?
L512: interobserver repeatability not mentioned in the introductory aim but it is here.
Conclusion: where is the refined list the study said it will proposed? this aim did not seems to be achieved as a list I assume there would be a kind of table or something of that nature
are the concluded variables the most repeatable among the observers? how do they benefit the ongoing welfare of rabbits during slaughter? what is the take home message after assessing the effectiveness of the methods?
Of the three objectives stated only one appears to have been achieved.
Again, this is an interesting paper and well-done to the authors for their hardwork.
Comments on the Quality of English LanguageShould be improved.
Author Response
See the document attached. Thanks for your revision!

Reviewer 3 Report
Comments and Suggestions for Authors
The subject of the article is very important, regarding animal welffare issues. However the description of materials and methods is too descriptive. The article is too long making it difficult to read. The diversity of results (1. Inter-observer repeatability of the animal-based indicators, 2. Relationship among animal-based indicators, 3. Relationship between key parameters and stunning efficiency and 4. Risk factor analysis, make discussion difficult.
Several references are missing and few in number

Author Response
Find the answer enclosed.

Reviewer 4 Report
Comments and Suggestions for Authors
Consciousness is usually related to being alive (breathing) and being aware of the environment. Unconsciousness of being alive and treathing but unable to respond to the environment, or feel.
Using breathing as an indicator of consciousness is therefore contentious, and needs further explanation. Certainly vocalisations would qualify as conscious in the normal use of the word, and there is justification given for the righting reflect, but what about blinking? Is this automatic ( ie unconscious) or not, some justification needed. So some clarity here in introduction and then emphasise the % occurence of "consciousness". One SH seemed to have a good stunner,, but it did not cause a tonic response, so recommended or not?
Certainly pr-stunning MUST morally result in no feeling thereafter, but increase clarity needed please, otherwise all OK although for someone marginally interested it is a bit long!
Author Response
Thank you for your thoughtful comment. Find the answer enclosed.

Round 2
Reviewer 1 Report
Comments and Suggestions for Authors
The statistical analysis, particularly the use of Fleiss’ kappa for inter-observer repeatability, is well-executed. However, the presentation of these results would benefit from greater clarity regarding the implications for practical applications in commercial slaughterhouses. For example, the relationship between κ values and their operational impact should be more explicitly discussed.
The manuscript acknowledges that the righting reflex is often confounded with spinal reflexes, such as the Lazarus sign. This raises significant concerns about the reliability of this indicator as a measure of consciousness. I strongly recommend that the authors reconsider the inclusion of the righting reflex as a valid animal-based indicator (ABI) unless its association with consciousness can be unequivocally established through more robust evidence.
While the authors claim that the slaughterhouses (SHs) selected represent a wide variety of operational setups, the inclusion criteria are not sufficiently detailed. This limitation should be explicitly acknowledged in the discussion, as it may affect the generalizability of the findings.
The manuscript would benefit from a deeper discussion of how the findings align with current welfare standards in the European Union and globally. Specifically, the ethical implications of using electrical stunning methods with variable effectiveness across SHs should be explored.
- Table 6 contains valuable data on inter-observer repeatability but lacks adequate labeling and formatting for easy interpretation. Please improve the table presentation.
- The description of the stunning devices (Section 2.2) is thorough but overly technical in places. Simplifying this section for a broader audience would enhance readability.
Author Response
Thank you for your time at revising the manuscript. Please, find the answer enclosed.
